# A DIFFERENTIABLE PHYSICAL SIMULATION FRAMEWORK FOR SOFT ROBOTS ON MULTIPLE-TASK LEARNING

## ABSTRACT

Learning multiple tasks is challenging for soft robots. Differentiable physics enables efficient gradient-based optimizations of neural network (NN) controllers for soft robot learning. However, existing work typically delivers NN controllers with limited capability and generalizability. We present a practical learning framework that outputs unified NN controllers capable of multiple tasks with significantly improved complexity and diversity. Our framework consists of a high-performance differentiable deformable bodies simulator supporting the material point method (MPM) and mass-spring systems, an automatic differentiation module that enables gradient-based optimizations, and a practical training module for soft robots on learning multiple locomotion tasks with a single NN controller. Using a unified NN controller trained in our framework, we demonstrate that users can interactively control soft robot locomotion and switch among multiple goals with specified velocity, height, and direction instructions. We evaluate our framework with multiple robot designs and challenging locomotion tasks. Experiments show that our learning framework, based on differentiable physics, delivers better results and converges much faster, compared with reinforcement learning frameworks. In addition, we successfully employed our framework on learning manipulation tasks, indicating the potential to extend our framework to tasks beyond locomotion.

## 1 INTRODUCTION

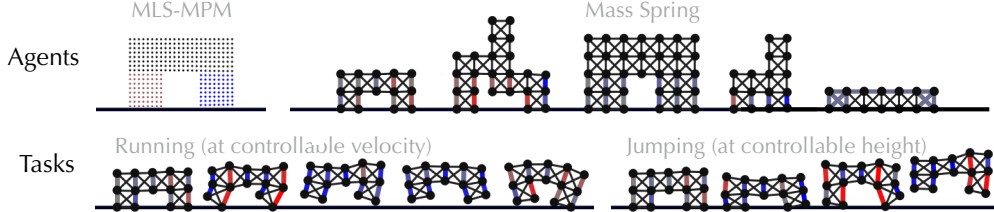

Figure 1: **Our learning system is robust and versatile, supporting various simulation methods, robot designs, and locomotion tasks with continuously controllable target velocity and heights.**

Differentiable physical simulators deliver accurate analytical gradients of physical simulations, opening up a promising stage for efficient soft robots control task learning via gradient descent. Existing research demonstrates that on simple tasks, learning systems via differentiable physics can effectively leverage simulation gradient information and converge orders of magnitude faster (see, e.g., de Avila Belbute-Peres et al. (2018); Hu et al. (2019)) than reinforcement learning.

However, the capability of existing differentiable physics based learning systems is relatively limited. Typically, optimized controllers can only achieve relatively simple single-goal tasks (e.g., moving in one specific direction, as in ChainQueen Hu et al. (2019)). Those learned controllers often have difficulty generalizing the task to a perturbed version.

In this work, we propose a learning framework for complex locomotion skill learning via differentiable physical simulation. The complexity of our tasks comes from two aspects: first, the degrees

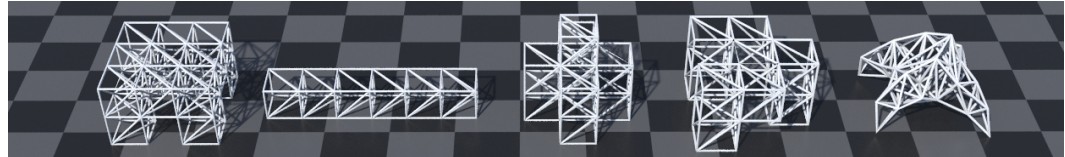

Figure 2: **3D robots collection.** Our framework supports various robot designs both in 2D and 3D.

of freedom of our soft robots are significantly higher compared with the rigid-body ones; second, the robots are expected to learn multiple skills at the same time. We systematically propose a suite of enhancements (Fig. 4) over existing training approaches (such as Hu et al. (2020); Huang et al. (2021)) to significantly improve the efficiency, robustness, and generalizability of learning systems based on differentiable physics.

In addition, we investigated the contributions of each enhancements to training in detail by a series of ablation studies. We also evaluated our framework through comparisons against Proximal Policy Optimization (PPO) Schulman et al. (2017), a state-of-the-art reinforcement learning algorithm. Results show that our system is simple yet effective and has a much-improved convergence rate compared to PPO.

We demonstrate the versatility of our framework via various physical simulation environments (mass-spring systems and the moving least squares material point method, MLS-MPM Hu et al. (2018)), robot designs (structured and irregular, Fig. 2), locomotion tasks (moving, jumping, turning around, all at a controllable velocity). To the best of our knowledge, this is the first framework which enables a single neural network can be trained via differentiable physics to achieve tasks of such complexity (Fig. 1). Our robot is simultaneously trained with multiple tasks, the learned skills can be continuously interpolated. For example, we can control both the velocity and height of an agent while it runs. The trained agent can also be controlled in real-time, enabling direct applications in soft robot control and rapid design. In addition, our framework is successfully applied to simple manipulation tasks (shown in Figure 10), which indicate that our framework has the potential to handle tasks beyond locomotion. The code and data of the framework will be open-source. We summarize our key contributions as follows:

- A GPU-accelerated high-performance differentiable physical simulation environment for deformable robot locomotion skills learning, which supports mass-spring systems and the material point method as dynamic simulation backends.

- A differentiable physics-based learning framework for soft robots on multiple locomotion tasks learning using a single NN controller. The trained controller can be used to control soft robots interactively, achieving multiple locomotion tasks.

- A systematic investigation of the key factors contributing to the differentiable physics based learning framework. We believe our investigation can inspire further research to explore the possibilities of differentiable physics for more complex robot learning tasks.

## 2 RELATED WORK

**Differentiable simulation.** Differentiable physical simulation is getting increasingly more attention in the learning community. Two families of methods exist: the first family uses neural networks to *approximate* physical simulators Battaglia et al. (2016); Chang et al. (2016); Mrowca et al. (2018); Li et al. (2018). Differentiating the approximating NNs then yields gradients of the (approximate) physical simulation. The second family is more accurate and direct: many of these methods using differentiable programming (specifically, reverse-mode automatic differentiation) to implement physical simulators Degrave et al. (2016); de Avila Belbute-Peres et al. (2018); Schenck & Fox (2018); Heiden et al. (2019); Hu et al. (2019; 2020); Huang et al. (2021). Automatic differentiation works well for explicit time integrators, but when it comes to implicit time integration, people often adopt the adjoint methods Bern et al. (2019); Geilinger et al. (2020), LCP de Avila Belbute-Peres et al. (2018) and QR decompositions Liang et al. (2019); Qiao et al. (2020). A smooth critic function is employed in Xu et al. (2021) to mitigate vanishing/exploding gradient issues; however, its application is confined to a single task. In this work, we leverage DiffTaichi Hu et al. (2020), an

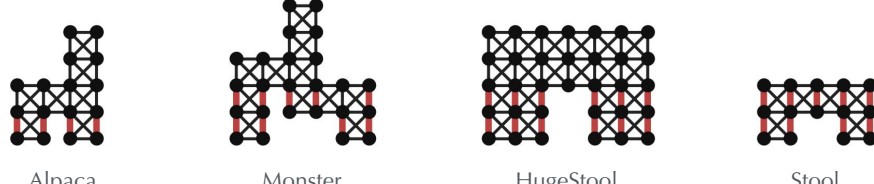

Figure 3: **2D robots collection**. The name of the robot is shown under the schematic.

automatic differentiation system, to create high-performance parallel differentiable simulators and the built-in NN controllers.

**Locomotion skill learning.** Producing physically plausible characters with various locomotion skills is a challenging problem. One classic approach is to manually design locomotion controllers subject to physics laws of motion in order to generate walking Ye & Liu (2010); Coros et al. (2010) or bicycling Tan et al. (2014) characters. These physically-based controllers often rely on a complex set of control parameters and are difficult to design and time-consuming to optimize. Another line of research suggests using data to control characters kinematically. Given a set of data clips, controllers can be made to select the best fit clip to properly react to certain situations Safonova & Hodgins (2007); Lee et al. (2010); Liu et al. (2010). These kinematic models use the motion clips to build a state machine and add transitions between similar frames in adjacent states. Although being able to produce higher quality motions than most simulation-based methods, the kinematic methods lack the ability to synthesize behaviors for unseen situations. With the recent advances in deep learning techniques, attempts have been explored to incorporate reinforcement learning (RL) into locomotion skill learning Peng et al. (2015); Liu & Hodgins (2017); Peng et al. (2018); Park et al. (2019). These modern controllers gain their effectiveness either from tracking high-quality reference motion clips or from cleverly designed rewards to imitate the reference. However, the exploration space for RL is usually prohibitively large to achieve complicated target motions. Carefully designed early termination strategies Ma et al. (2021); Won et al. (2020), better optimization methods Yang & Yin (2021), and adversarial RL schemes Peng et al. (2021) enable these RL-based methods to achieve richer behaviors. However, it is still time-consuming to get a well-trained RL model in complex tasks such as soft robot control. Our method, on the other hand, leverages the differentiable simulation framework and can achieve much better convergence behavior compared to RL-based methods.

## 3 DIFFERENTIABLE SIMULATION ENVIRONMENTS

### 3.1 SIMULATION CONFIGURATIONS

Our learning framework supports multiple types of differentiable physically-based deformable body simulators. We use the mass-spring systems and the moving least squares material point method (MLS-MPM) as our simulation environments. The design choices for each environment are shown below:

**Mass-spring systems** We adopt the classic Hookean spring model to represent the elastic force and use dashpot damping Baraff & Witkin (1998) as the damping force to simulate our mass-spring system. We found that the drag damping model used by Hu et al. Hu et al. (2020) damps the gradient of the NN controller as well while the dashpot damping model is able to generate vividly changing gradients which makes our agent more flexible.

**Material point methods** We use MLS-MPM Hu et al. (2018) as our material point method simulator. We further applied the affine particle-in-cell method Jiang et al. (2015) to reduce the artificial damping. Due to the nature of MPM as a hybrid Lagrangian-Eulerian method, it always requires a background grid during the simulation. Naive MPM implementations fix the position of the background grid, hence limit the moving range of the agent. We apply a dynamic background grid that follows the agent all the time to overcome this problem.

## 3.2 ROBOT STRUCTURE AND ACTUATION MODE

Our framework supports a variety of differently shaped agents as shown in Fig. 2 and 3. Our actuation signal exerts forces along with the "muscle" directions of the simulated agents. This magnitude of the is limited within the range $[-1, +1]$, where its sign determines whether an agent wants to contract (negative sign) or relax (positive sign) a muscle and its absolute value determines the magnitude of the force.

In mass-spring systems, the muscle directions are represented by the spring directions, we change the rest-length of springs to generate forces. Given a spring with two mass points $\mathbf{x}_0$ and $\mathbf{x}_1$ and rest-length $l_0$. The displacement (vector) $\mathbf{d}$ is defined by equation: $\mathbf{d} = (||\mathbf{x}_0 - \mathbf{x}_1|| - l_0)\frac{\mathbf{x}_0 - \mathbf{x}_1}{||\mathbf{x}_0 - \mathbf{x}_1||}$. Taking an analogy to the muscle contraction model, we set the rest-length $l_0$ to desired length according to the actuation signal from the NN controller to control the displacement and the force thereby. The actuation signal is used to scale the rest-length up to some limit (usually at $20\%$). In addition, only activated springs, i.e. actuators (springs marked red or blue shown in Fig. 1), are able to generate forces according to control signals.

In material point methods, the muscle directions are always along the vertical axis in material space. In our implementation, we simply modify the Cauchy stress in the vertical direction of material space to apply this force. Again, the force is scaled to a user-defined bound which can be adjusted for different simulations.

# 4 LEARNING FRAMEWORK

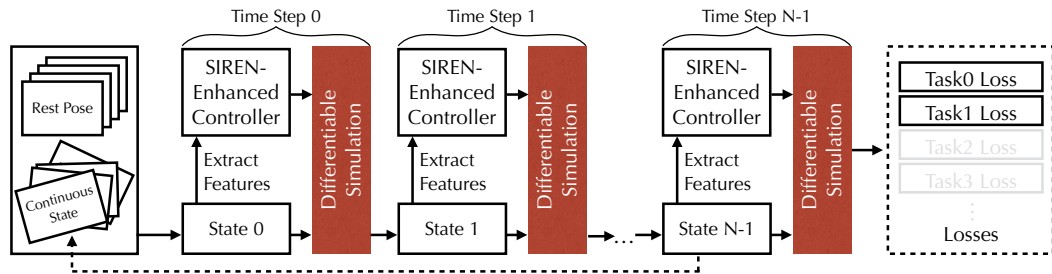

Figure 4: **Framework overview.** Simulation instances are batched and executed in parallel on GPUs. The whole system is end-to-end differentiable, and we use gradient-based algorithms to optimize the neural network controller weights. Each time step involves evaluating a NN controller inspired by SIREN Sitzmann et al. (2020), and a differentiable simulator time integration implemented using DiffTaichi Hu et al. (2020). Simulation states are fed back to the initial state pool to improve the richness of training sets and thereby the generalization ability of the resulted NN controller. A tailored loss function (as shown in section 4.2) is designed for each task.

In this section, we describe our training framework based on differentiable physics (Fig. 4). In summary, we develop a differentiable simulator with an embedded NN controller. In each time step, the program performs an NN controller inference and a differentiable simulator time integration. After a few hundred time steps, the program back-propagates gradients end-to-end via reverse-mode automatic differentiation, and updates the weights of the neural network using the Adam optimizer Kingma & Ba (2014).

## 4.1 TASK REPRESENTATION

Given an agent $A$, its position and corresponding velocity are denoted as $\mathbf{x} = (x, y, z)$ and $\mathbf{v} = (u, v, w)$. We presents three tasks (running, jumping and crawling) for 2D agents, two (running and rotating) for 3D. In a whole simulation with total steps $\mathcal{T}$, an agent is expected to achieve all goals in a goal sequence $\mathcal{G} = \{\mathbf{G}_1, \mathbf{G}_2...\mathbf{G}_n | n\mathcal{P} = \mathcal{T}\}$, where $\mathcal{P}$ is the time period for an agent to achieve a goal, $n$ is the number of goals . For each goal $\mathbf{G}$, there are multiple tasks in it. Each task is encoded using a target value $g$, which instructs the controller to drive the agent. For example, $\mathbf{G} = \{g_v, g_h\}$ represent performing running and jumping simultaneously, the target velocity and height are $g_v$ and $g_h$. The agent is expected to switch between multiple goals during one simulation.

**Running** The running task is defined as a velocity tracking problem. Given a time step $t$, the center of mass $\mathbf{c}$ is used to represent the position of the agent, its velocity is defined as the averaged velocity $\tilde{\mathbf{v}}$ over a time period $\mathcal{P}_r$, ($\mathcal{P}_r < \mathcal{P}$). Then the agent is expected to achieve a velocity $\tilde{\mathbf{v}}$ close to the target velocity $g_v$.

$$\tilde{\mathbf{v}}(t) = \frac{1}{\mathcal{P}_r} \left( \mathbf{c}(t) - \mathbf{c}(t - \mathcal{P}_r) \right) \tag{1}$$

**Rotation** In a 3D space, an agent is additionally expected to control its orientation on a plane, i.e., rotate. The rotation matrix $\mathbf{R}$ along X-Z plane is defined as

$$\mathbf{R} = \begin{bmatrix} \cos(\mathcal{P}_r g_\omega) & 0 & -\sin(\mathcal{P}_r g_\omega) \\ 0 & 1 & 0 \\ \sin(\mathcal{P}_r g_\omega) & 0 & \cos(\mathcal{P}_r g_\omega) \end{bmatrix}, \tag{2}$$

where $g_\omega$ is the target angular velocity assembled in input features.

**Jumping** The jumping task is defined as reaching a given height $g_h$ in a time period $\mathcal{P}$. Due to the gravity, it is not possible for a running robot to stay away from the ground. Therefore, the jumping height $\tilde{h}_h(t)$ of an agent is defined by the maximum vertical position of its lowest point in a certain period $\mathcal{P}$. To be more formally, given $\mathbf{S}$ a set of nodes (or points) that constitute the agent, $h$ is a function that can extract the vertical position given a node (or point), the jumping height over the $n$-th period is defined as

$$\tilde{h}_h(n) = \max_{t \in [0, \mathcal{P}]} \min_{s \in \mathbf{S}} h(s, t + n\mathcal{P}) \tag{3}$$

where $n$ is the index of the period in a whole simulation.

**Crawling** The crawling task is defined as lowering the highest point as much as possible. The target of crawling $g_c$ is defined as an indicator function. The agent is controlled to crawl if $g_c = 1$ otherwise if $g_c = 0$. Since crawling is a status that needs to be maintained, the crawling height is defined as the vertical position of its highest point at each time step.

$$\tilde{h}_c(t) = \max_{s \in \mathbf{S}} h(s, t) \tag{4}$$

## 4.2 LOSS FUNCTIONS

Consider a locomotion task where an agent is instructed to move at a specified velocity, we think a proper loss function should satisfy the following requirements:

1. **Periodicity.** The agent is expected to move periodically following a predefined cyclic activation signal. Therefore, the loss function should encourage periodic motions.

2. **Delayed evaluation.** Due to inertia, it takes time for the agent to start running or adjust running velocity. Therefore, an ideal loss function should take this delay into consideration.

3. **Fluctuation tolerance.** During a time period for an robot to achieve a goal, requiring the center of mass to move at a *constant* velocity at each time step may induce highly fluctuated losses. To avoid that, our loss function should be smooth during one period.

**Task loss** For each task, we defined a tailored loss shown in equation 5 according to the requirements above. The loss for running $\mathcal{L}_v$ is defined as the accumulation of the difference between the target and the agent's velocity from start to current time step. For jumping, we define a sparse loss $\mathcal{L}_h$, which only evaluate once for each period. The crawling loss $\mathcal{L}_c$ is defined as the accumulation

of the crawling height if the loss is applicable.

$$\mathcal{L} = \lambda_v \underbrace{\sum_{n \in \mathcal{T}} \sum_{t=\mathcal{P}_r}^{\mathcal{P}} (\tilde{\mathbf{v}}(t) - g_v(n))^2}_{\mathcal{L}_v} +$$

$$\lambda_h \underbrace{\sum_{n \in \mathcal{T}} (\tilde{h}_h(n) - g_h(n))^2}_{\mathcal{L}_h} +$$

$$\lambda_c \underbrace{\sum_{n \in \mathcal{T}} \sum_{t=0}^{\mathcal{P}} g_c(n) \tilde{h}_c(t + n\mathcal{P})}_{\mathcal{L}_c} \tag{5}$$

For 3D cases, the additional rotation loss $\mathcal{L}_r$ is defined as the accumulation of centralized point-wise distance, i.e., the relative distance to the center of mass, between current position and target rotated position

$$\mathcal{L}_r = \lambda_r \sum_{n \in \mathcal{T}} \sum_{t=\mathcal{P}_r}^{\mathcal{P}} \sum_{s \in \mathbf{S}} (\bar{\mathbf{x}}(t) - \mathbf{R}\bar{\mathbf{x}}(t - \mathcal{P}_r))^2 \tag{6}$$

where $\bar{\mathbf{x}}(t) = \mathbf{s}(t) - \mathbf{c}(t)$ is the centralized point-wise distance. The $\lambda_v, \lambda_h, \lambda_c, \lambda_r$ are weights for losses of dfferent tasks. Our loss function accumulates the contributions of all steps in a "sliding window". This prevents the loss value being too small as well as the gradient vanishing problem. An accumulated loss also helps alleviate the velocity loss fluctuation during training.

**Regularization**   Since the agents may keep shaking when no target values are given, we introduce an actuation loss as a regularization term. The intuition is to penalize comparatively large actuations when small target value is given:

$$\mathcal{L}_a = \sum_{n \in \mathcal{T}} \sum_{t \in [0, \mathcal{P}]} (\mathcal{A}(t) - \mu ||g_v(n)||)^2 \tag{7}$$

where $\mathcal{A}(t)$ is the actuation output of NN, $\mu$ is a normalizer.

### 4.3 NETWORK ARCHITECTURE

We use two fully connected (FC) layers with sine activation function as the neural network. The network input vector consists of three parts:

1. **Periodic control signal** of the same period with different phases, to encourage periodic actuation. In this work, we use sin waves as the signal.
2. **State feature vector** is a representation of the current state of the agent. The positions and velocities of all the vertices of the simulated object are used as the input feature. To ensure the property translation-invariant and rotation-invariant, we use the the relative distance to the center of mass to remove global translation and rotation information.
3. **Targets** that encode the task information such as running velocity, jumping height, and orientation for 3D agents.

### 4.4 TRAINING

In the training process, a simulation with 1000 steps is performed during one training iteration. The duration of the simulation is divided into several periods $\mathcal{P}$ as mentioned in the section 4.1. For each $\mathcal{P}$, a goal $\mathbf{G}$ contains several tasks is assigned. For example, $\mathbf{G} = \{g_v, g_h\}$ represents that the agent is expected to perform both running and jumping in this period. The goal and its target values are generated randomly in a uniform distribution. They are kept fixed inside one period $\mathcal{P}$, but varied between different periods. The network is trained with a batch size of 32. We train our models on NVidia RTX2060 GPU, the maximum training iteration is set to 10000, while the it usually converges in less than 5000 iterations within half an hour.

# 5 EXPERIMENTS AND ANALYSIS

In this section, we evaluate our framework and learning algorithm from various aspects.

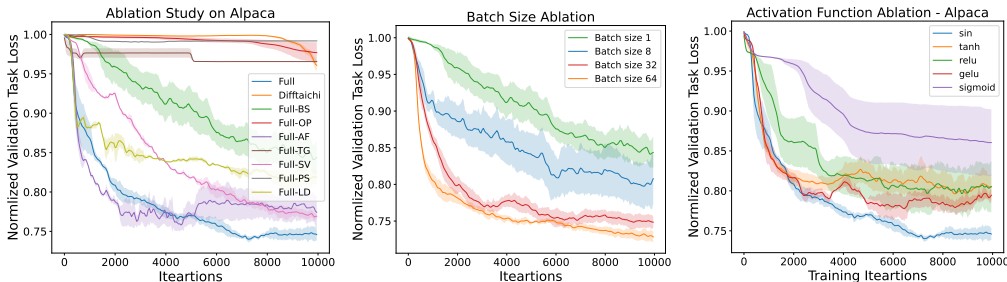

Figure 5: **Summary of the ablation study (left).** Here we show the summary of ablation study on robot *Alpaca*.The abbreviations under training setting represent name of components to remove. To be more specific, OP: Optimizer, AF: Activation Function, BS: Batch Size, PS: Periodic Signal, SV: State Vector, TV: Targets, LD: Loss Design. Difftaichi is an implementation of Hu et al. (2020), enhanced with our loss design. It can be observed that the *Full* method achieves the best performance on *Task* loss among all models. Each iteration indicates one training iteration, which is composed of 1000 steps of physical simulation and one step of update on weights of the controller. The result show that the *Full* method achieves the best performance. **Ablation study of batch size (middle). Ablation study of activation function (right).** For more results of ablation studies on other robots, please check the appendix.

## 5.1 FRICTION MODEL

In real world, robots running is driven by static friction between "feet" and ground. We train robots with different contact models in our experiments. With either zero friction or a fully sticky surface, where only velocity in normal direction is kept and all other components are projected to zero after contact, the robot tends to stuck shown in the middle of Fig.6. To overcome this issue, we adopt the classic slip with friction model from physical simulation which involves both kinetic and static friction. The friction force of the two contact modes are determined by the pressure force applied on the contact surface and a given friction coefficient.

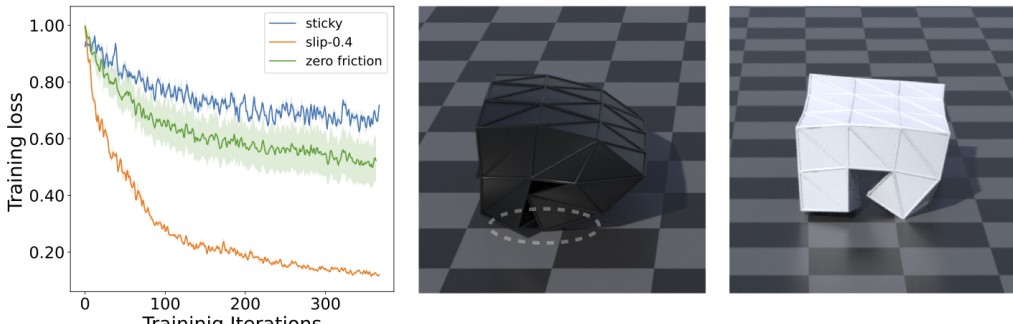

Figure 6: **Left:** Training loss for zero friction, slip and sticky contact models. The 0.4 is the value of the friction coefficient. **Middle:** Robot suffers from sticky surface and cannot move further. **Right:** Robot moves smoothly towards right. Please see the supplemental video for a visual comparison between slip and sticky friction models.

## 5.2 DIFFERENTIABLE PHYSICS GRADIENT ANALYSIS

One concern in differentiable physics based learning is the that the gradient may explode during the back-propagation through long simulation steps. Here we visualize the distribution of gradients norm of a whole training process in Fig. 7. It can be observed that most values of the gradients are distributed inside the interval [-10, 10] in log scale, which indicates that our learning approach provides stable gradients during training.

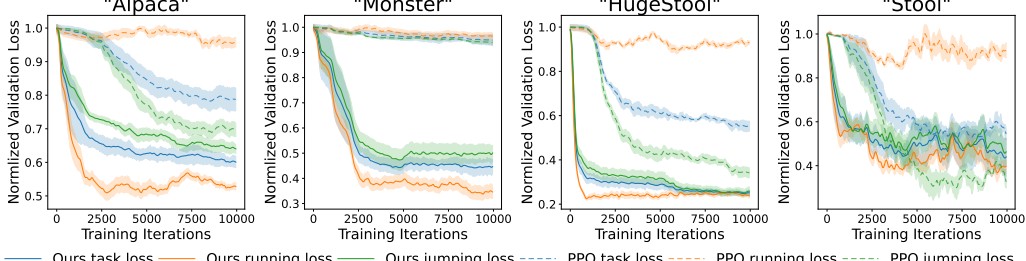

Figure 7: **Gradient Analysis.** The plots show the gradient distribution of different robot. The x-axis represent the sum of gradients norm. The values are drawn in log scale for a clear visualization. We record the sum of gradients norm of 10000 iterations for a whole training process. For each robot, the experiments are performed at least three times, i.e., there are at least 30000 gradient samples for one robot.

## 5.3 LEARNING TECHNIQUES

we systematically investigate the key factors of the learning techniques that contribute to the training, evaluate our framework through ablation studies. All experiments in the ablation studies are performed at least three times. The results of ablation studies are summarized in Fig. 5. In validation, we test all the trained robots on a fixed set of tasks, whose target values are uniformly sampled by interpolating between the lower and upper bound of all targets appeared in training.

**Batching**   We adopt batching into differentiable physics and perform a series of experiments as shown in the middle of Fig. 5 to verify its effectiveness, with batch size 1, 8, 32, and 64. It shows that a batch size of 32 gives the best trade-off between performance improvement and computation cost.

**Loss design**   We compare our tailored loss function with a naive loss function. The naive loss function defines the velocity as the position difference of the center of mass between consecutive time steps, i.e., requiring the robot to move at a constant velocity at each time step. The experiments show that the robot under the naive loss design can not be properly trained to run, i.e., there are almost no progress in running loss dropping.

**Activation function**   To validate the effectiveness of our $sin$ activation function, we replaced it for hidden layer with various popular activation functions $tanh$, $relu$, $gelu$ and $sigmoid$. The results in the right of Fig.5 show that using $sin$ as activation functions gives the best performance among all three losses. For more results of activation functions ablation studies, please check the table 6 in appendix.

## 5.4 BENCHMARK AGAINST REINFORCEMENT LEARNING

Figure 8: **Comparison on different robots.** Both our method and PPO run on GPUs. The solid and dashed lines show the validation loss of our method and PPO, respectively. The robot design is also shown in the figure. Springs marked in red are actuators.

We implement the standard proximal policy optimization (PPO) benchmark in multiple goals settings as shown in Fig. 8. The PPO robot can make progress in single goal learning and learn certain locomotion patterns to move toward the goal. Yet as the task gets complicated, PPO often gets trapped in the local minima. We try our best to fine-tune the PPO hyper-parameters, but still find it tends to overfit to one goal but fails to find a balance between different goals. It can be observed in Fig. 8 where the robots trained by PPO can learn to jump but struggle to run.

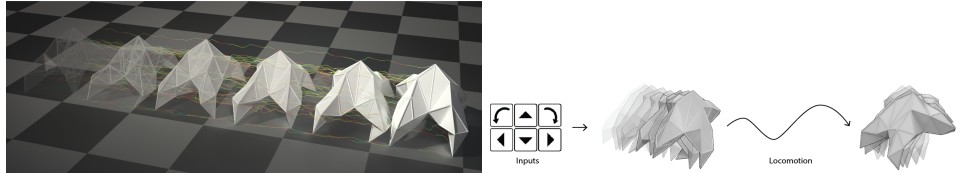

Figure 9: **Top:** the trajectory of a walking quadruped robot under user control. **Bottom:** Our system allows the user to control the robot's motion interactively in real-time. Please see the supplemental video for results in detail.

## 5.5 INTERACTIVE REAL-TIME CONTROL

Smoothly switching between different tasks is a key advantage of our approach. In interactive settings, the target values are provided by the user inputs, users can smoothly control the robot's motion in real-time via input devices such as a keyboard as shown in Fig. 9. Please refer to our supplemental video for more details.

## 5.6 EXTENSIBILITY OF THE FRAMEWORK

Our framework is not limited to locomotion tasks learning. Here we provide two simple manipulation learning tasks trained on our framework. In these tasks, the robot is expected to manipulate an object (marked in purple) to hit a target point (marked in green). We designed two scenarios based on MPM: *Juggle* and *Dribbble and Shot* shown in Fig.10.

**Juggle**  In this scenario, the target point appears in the sky above the the robot. The robot is expected to 'juggle' the object to the target point.

**Dribble and Shot**  In this scenario, the target point keeps moving toward right. The robot is expected to carry the object and shot it to the target point.

The experiments indicate that our framework has the extensibility to tasks beyond locomotion.

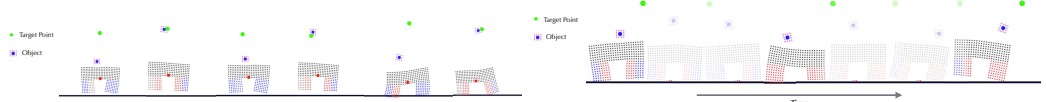

Figure 10: **Manipulation tasks showcase.** The upper plot shows the snapshots of scenario 'Juggle' and the lower one shows that of the 'Dribble and Shot'. The visual results are also shown in the supplemental video.

## 6 DISCUSSION

In this paper, we have presented a differentiable physics based learning framework which enables users to flexibly design soft robots and to teach them multiple locomotion skills. It consists of a high-performance differentiable deformable bodies simulator, an automatic diffrentiation module and a tailored learning algorithm for soft robot on multiple-task locomotion skills learning with single controller. The algorithm shows advantages in both training performance comparing to PPO. and convergence efficiency on multiple-task The key factors of learning techniques contributing to the training are systematically investigated. The trained robots can be manipulated to smoothly switch locomotion tasks such as running, jumping, and crawling with different speeds and orientations, interactively in real-time. In addition, we successfully applied the proposed method to simple manipulation tasks, which indicates the framework has the extensibility to tasks beyond locomotion.

There are also some limitations. Difficulty of the learning tasks depends on robot design and physical parameters. The training performance may be degraded with improper robot designs. For instance, in our stool case where unnecessary actuators on its body are allowed, it is more difficult to achieve good results. Offloading the physical parameter tuning and robot designing to an automatic pipeline will be an interesting future research direction.

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

# A  APPENDIX

## A.1  REINFORCEMENT LEARNING SETTING

**Environment.**  We use the open-source implementation of PPO (Kostrikov (2018)) in our environments. Part of important hyper-parameters are listed in the table 1.

Table 1: PPO hyper-parameters

| Parameter | Values |
|-----------|--------|
| learning rate | $2.5e-4$ |
| entropy coef | 0.01 |
| value loss coef | 0.5 |
| number of processes | 8 |
| number of simulation steps. | 1000 |

**Reward Design.**  We design a reward function for PPO based on our loss functions. Recall the velocity part of equation 5. We define the total velocity reward as:

$$\mathcal{R}_v = \lambda_v \sum_{n \in \mathcal{T}} \sum_{t \in [\mathcal{P}_r, \mathcal{P}]} g_v(n)^2 - (\tilde{\mathbf{v}}(t) - g_v(n))^2 \tag{8}$$

If we directly split the reward into each time steps by $t$, the rewards of first $\mathcal{P}_r$ time steps would be zero and it is too difficult for PPO to learn the policy. Therefore, we modified the reward function to tackle this issue. By equation 1, we have

$$\mathcal{R}_v = \lambda_v \sum_{n \in \mathcal{T}} \sum_{t \in [\mathcal{P}_r, \mathcal{P}]} g_v(n)^2 - \left[ \frac{1}{\mathcal{P}_r} \left( \mathbf{c}(t) - \mathbf{c}(t - \mathcal{P}_r) \right) - g_v(n) \right]^2 \tag{9}$$

$$= \frac{\lambda_v}{\mathcal{P}_r^2} \sum_{n \in \mathcal{T}} \sum_{t \in [\mathcal{P}_r, \mathcal{P}]} [\mathcal{P}_r g_v(n)]^2 - [\mathbf{c}(t) - (\mathbf{c}(t - \mathcal{P}_r) + \mathcal{P}_r g_v(n))]^2 . \tag{10}$$

We can define a function $f_n(t', t) = [\mathcal{P}_r g_v(n)]^2 - [\mathbf{c}(t') - (\mathbf{c}(t - \mathcal{P}_r) + \mathcal{P}_r g_v(n))]^2$ and substitute it back to equation 10. The reward can be re-written as

$$\mathcal{R}_v = \frac{\lambda_v}{\mathcal{P}_r^2} \sum_{n \in \mathcal{T}} \sum_{t \in [\mathcal{P}_r, \mathcal{P}]} f_n(t, t) \tag{11}$$

Refer to that for any $t \in \mathcal{P}$, we have $f_n(t - \mathcal{P}_r, t) = 0$, which means

$$\mathcal{R}_v = \frac{\lambda_v}{\mathcal{P}_r^2} \sum_{n \in \mathcal{T}} \sum_{t \in [\mathcal{P}_r, \mathcal{P}]} \sum_{\Delta t \in [0, \mathcal{P}_r)} f_n(t - \Delta t, t) - f_n(t - \Delta t - 1, t) \tag{12}$$

Let $t' = t - \Delta t$

$$\mathcal{R}_v = \frac{\lambda_v}{\mathcal{P}_r^2} \sum_{n \in \mathcal{T}} \sum_{t' \in [0, \mathcal{P}]} \sum_{\Delta t = 0}^{\min(t', \mathcal{P}_r)} f_n(t', t' + \Delta t) - f_n(t' - 1, t' + \Delta t) \tag{13}$$

$$= \frac{\lambda_v}{\mathcal{P}_r^2} \sum_{n \in \mathcal{T}} \sum_{t' \in [0, \mathcal{P}]} \sum_{\Delta t = 0}^{\min(t', \mathcal{P}_r)} - [\mathbf{c}(t') - (\mathbf{c}(t' + \Delta t - \mathcal{P}_r) + \mathcal{P}_r g_v(n))]^2 + $$
$$[\mathbf{c}(t' - 1) - (\mathbf{c}(t' + \Delta t - \mathcal{P}_r) + \mathcal{P}_r g_v(n))]^2 \tag{14}$$

So we can split the whole reward into each time step $t$ as:

$$\mathcal{R}_v(t) = \sum_{\Delta t=0}^{\min(t, \mathcal{P}_r)} [\mathbf{c}(t-1) - (\mathbf{c}(t + \Delta t - \mathcal{P}_r) + \mathcal{P}_r g_v(n))]^2 - $$
$$[\mathbf{c}(t) - (\mathbf{c}(t + \Delta t - \mathcal{P}_r) + \mathcal{P}_r g_v(n))]^2 \quad (15)$$

Recall that the goal of a jumping task is to reach a specified height goal $g_h$, the intuition for jumping reward is to encourage the agent to improve its maximum height until reaching the height.

$$\mathcal{R}_h(t) = \lambda_h \sum_{n \in \mathcal{T}} (\min_{s \in \mathbf{S}} h(s, t + n\mathcal{P}) - g_h(n))^2 - (\tilde{h}_h(n) - g_h(n))^2 \quad (16)$$

where the $\min_{s \in \mathbf{S}} h(s, t + n\mathcal{P})$ is the agent height at current time step as mentioned in 4.1 and $\tilde{h}_h(n)$ is the max height record during one task period.

## A.2 NETWORK ARCHITECTURE

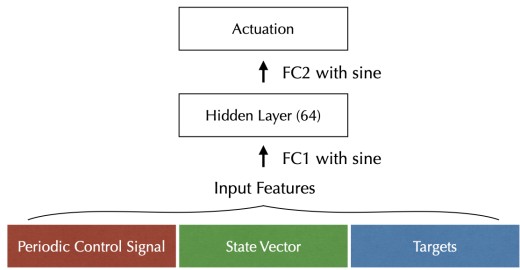

Figure 11: **Network architecture.**

Our network architecture is shown in Fig. 11. It is a two layers fully connected network with a 64 channels hidden layer. The number of input and output channels varies according to different agent designs.

## A.3 DETAILED RESULTS ON FURTHER EXPERIMENTS

Here we show more detailed results on further experiments, including ablation studies on different agents, validation loss on different tasks for agents and more network weights visualization results.

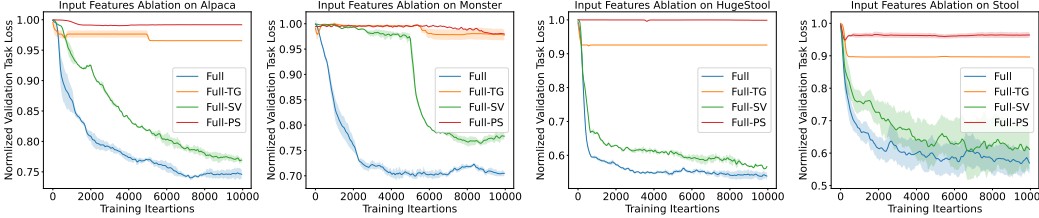

Figure 12: **Input features ablation study.** Full-PS, Full-SV and Full-TG indicate the models trained without Periodic Signal, State Vector or Targets respectively.

| Name | Training Setting | | | | | | | Norm. Valid. Loss (Alpaca) | | |
|---|---|---|---|---|---|---|---|---|---|---|
| | OP | AF | BS | PS | SV | TG | LD | Run. | Jump. | Task |
| Full | Adam | sin | 32 | ✓ | ✓ | ✓ | ✓ | 0.58±0.01 | 0.79±0.01 | 0.74±0.01 |
| Difftaichi* | SGD | tanh | 1 | ✓ | ✓ | ✓ | ✓ | 0.98±0.00 | 0.95±0.01 | 0.96±0.01 |
| Full-BS | Adam | sin | 1 | ✓ | ✓ | ✓ | ✓ | 0.82±0.02 | 0.85±0.01 | 0.84±0.02 |
| Full-OP | SGD | sin | 32 | ✓ | ✓ | ✓ | ✓ | 0.97±0.01 | 0.98±0.01 | 0.98±0.01 |
| Full-AF | Adam | tanh | 32 | ✓ | ✓ | ✓ | ✓ | 0.67±0.02 | 0.81±0.01 | 0.77±0.02 |
| Full-PS | Adam | sin | 32 | ✗ | ✓ | ✓ | ✓ | 0.97±0.00 | 0.99±0.00 | 0.99±0.00 |
| Full-SV | Adam | sin | 32 | ✓ | ✗ | ✓ | ✓ | 0.54±0.01 | 0.84±0.01 | 0.77±0.01 |
| Full-TG | Adam | sin | 32 | ✓ | ✓ | ✗ | ✓ | 0.99±0.00 | 0.95±0.00 | 0.97±0.00 |
| Full-LD | Adam | sin | 32 | ✓ | ✓ | ✓ | ✗ | 0.99±0.01 | 0.74±0.01 | 0.82±0.01 |

Table 2: **Ablation summary on agent *Alpaca*.** In this table, we show the *Task*, *Run* and *Jump* validation loss of proposed method and its ablated versions. *Full* represents the proposed method. The abbreviations under training setting represent name of components to remove. To be more specific, OP: Optimizer, AF: Activation Function, BS: Batch Size, PS: Periodic Signal, SV: State Vector, TV: Targets, LD: Loss Design. *Difftaichi is an implementation of Hu et al. (2020), enhanced with our loss design. It can be observed that the *Full* method achieves the best performance on *Task* loss among all models.

Table 3: **Ablation Summary on *Monster*.**

| Name | Training Setting | | | | | | | Norm. Valid. Loss (Monster) | | |
|---|---|---|---|---|---|---|---|---|---|---|
| | OP | AF | BS | PS | SV | TG | LD | Run. | Jump. | Task |
| Full | Adam | sin | 32 | ✓ | ✓ | ✓ | ✓ | 0.46±0.01 | 0.77±0.00 | 0.71±0.01 |
| Full-OP | SGD | sin | 32 | ✓ | ✓ | ✓ | ✓ | 0.99±0.01 | 0.99±0.00 | 0.99±0.01 |
| Full-AF | Adam | tanh | 32 | ✓ | ✓ | ✓ | ✓ | 0.65±0.01 | 0.84±0.02 | 0.79±0.02 |
| Full-PS | Adam | sin | 32 | ✗ | ✓ | ✓ | ✓ | 0.93±0.00 | 0.99±0.00 | 0.98±0.00 |
| Full-SV | Adam | sin | 32 | ✓ | ✗ | ✓ | ✓ | 0.51±0.01 | 0.86±0.01 | 0.78±0.01 |
| Full-TG | Adam | sin | 32 | ✓ | ✓ | ✗ | ✓ | 0.99±0.00 | 0.97±0.01 | 0.98±0.01 |
| Full-LD | Adam | sin | 32 | ✓ | ✓ | ✓ | ✗ | 0.99±0.00 | - | 0.90±0.06 |

Table 4: **Ablation Summary on *HugeStool*.**

| Name | Training Setting | | | | | | | Norm. Valid. Loss (HugeStool) | | |
|---|---|---|---|---|---|---|---|---|---|---|
| | OP | AF | BS | PS | SV | TG | LD | Run. | Jump. | Task |
| Full | Adam | sin | 32 | ✓ | ✓ | ✓ | ✓ | 0.40±0.01 | 0.58±0.01 | 0.53±0.01 |
| Full-OP | SGD | sin | 32 | ✓ | ✓ | ✓ | ✓ | 0.97±0.01 | 0.96±0.01 | 0.96±0.01 |
| Full-AF | Adam | tanh | 32 | ✓ | ✓ | ✓ | ✓ | 0.49±0.01 | 0.59±0.01 | 0.56±0.01 |
| Full-PS | Adam | sin | 32 | ✗ | ✓ | ✓ | ✓ | 0.99±0.00 | 0.99±0.00 | 0.99±0.00 |
| Full-SV | Adam | sin | 32 | ✓ | ✗ | ✓ | ✓ | 0.40±0.01 | 0.62±0.01 | 0.57±0.00 |
| Full-TG | Adam | sin | 32 | ✓ | ✓ | ✗ | ✓ | 0.99±0.00 | 0.90±0.00 | 0.93±0.00 |
| Full-LD | Adam | sin | 32 | ✓ | ✓ | ✓ | ✗ | 0.99±0.01 | - | 0.70±0.01 |

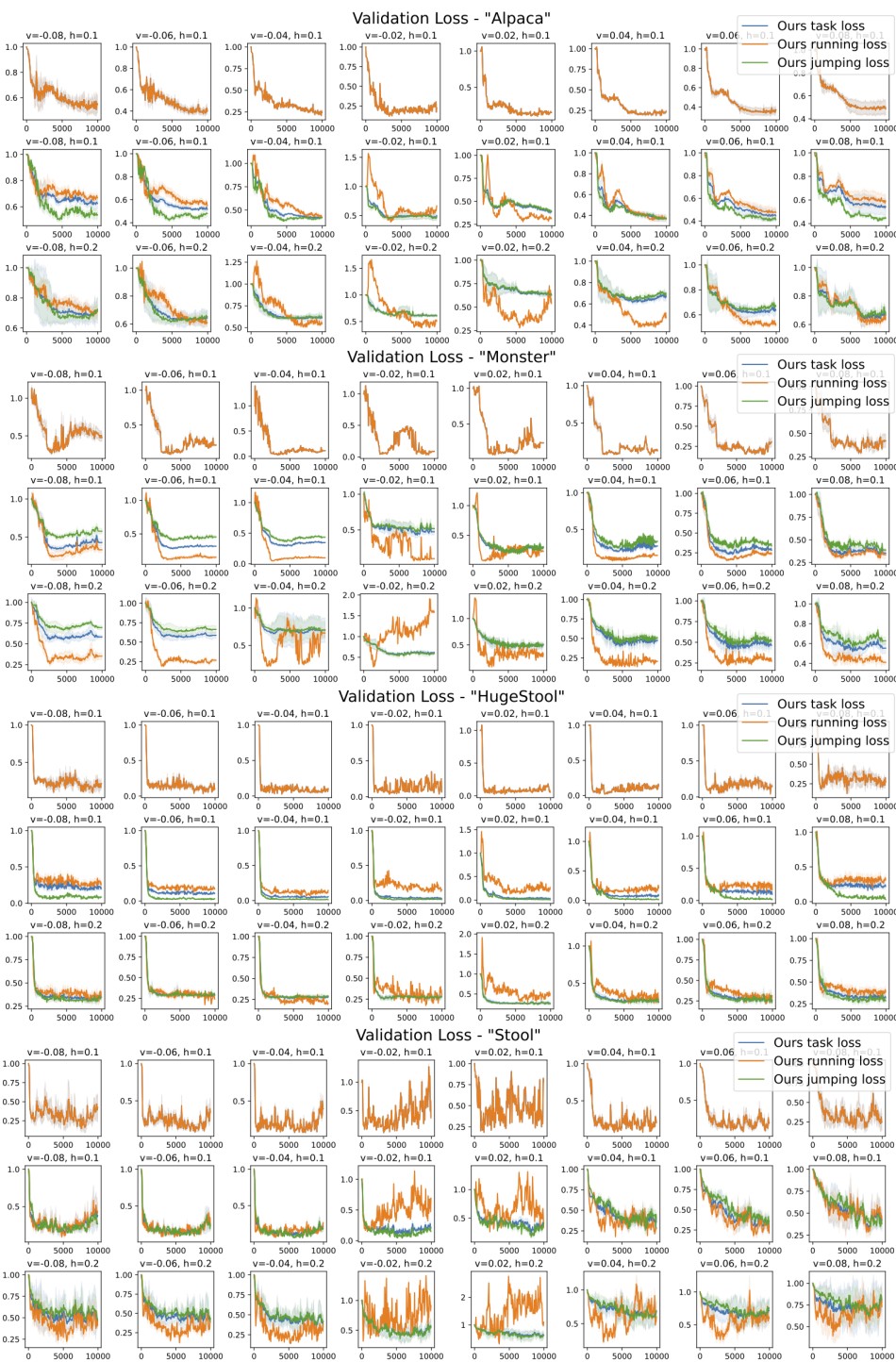

Figure 13: **Validation loss on different agents and targets.** The plot shows the validation loss on different targets combinations for agents. The target velocity ranges from -0.08 to 0.08 and the target heights are 0.1, 0.15 and 0.20. Each subplot shows the task, running and jumping loss for one agent given a pair of specified target velocity and height.

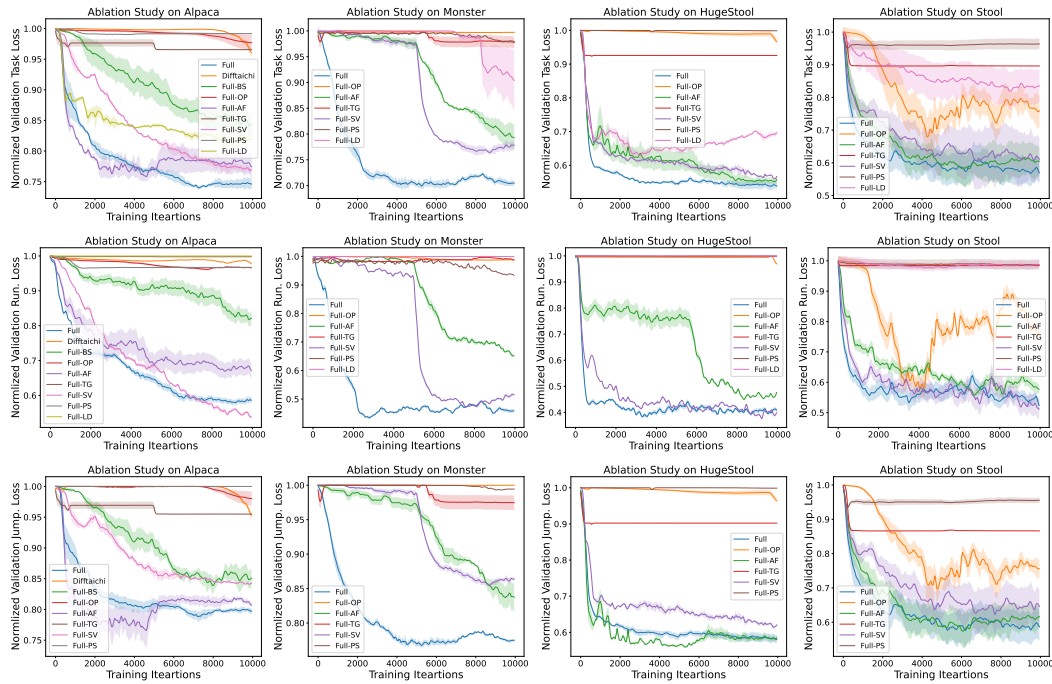

Figure 14: **Summary of the ablation study for agents.** The subplots show the normalized validation loss for task (weighted summation of different losses), running and jumping from top to bottom row. The subplots in different columns show the results of different agents.

Table 5: **Ablation Summary on *Stool*.**

| Name | Training Setting | | | | | | | Norm. Valid. Loss (Stool) | | |
|---|---|---|---|---|---|---|---|---|---|---|
| | OP | AF | BS | PS | SV | TG | LD | Run. | Jump. | Task |
| Full | Adam | sin | 32 | ✓ | ✓ | ✓ | ✓ | 0.50±0.01 | 0.58±0.04 | 0.56±0.03 |
| Full-OP | SGD | sin | 32 | ✓ | ✓ | ✓ | ✓ | 0.78±0.02 | 0.76±0.02 | 0.76±0.02 |
| Full-AF | Adam | tanh | 32 | ✓ | ✓ | ✓ | ✓ | 0.60±0.01 | 0.63±0.03 | 0.62±0.04 |
| Full-PS | Adam | sin | 32 | ✗ | ✓ | ✓ | ✓ | 0.98±0.01 | 0.96±0.00 | 0.96±0.02 |
| Full-SV | Adam | sin | 32 | ✓ | ✗ | ✓ | ✓ | 0.50±0.03 | 0.64±0.05 | 0.61±0.06 |
| Full-TG | Adam | sin | 32 | ✓ | ✓ | ✗ | ✓ | 0.98±0.00 | 0.87±0.01 | 0.89±0.01 |
| Full-LD | Adam | sin | 32 | ✓ | ✓ | ✓ | ✗ | 0.99±0.01 | - | 0.83±0.05 |

| Output \ Hidden | sin | tanh | relu | gelu | sigmoid |
|---|---|---|---|---|---|
| sin | **0.745** | 0.771 | 0.765 | 0.803 | 0.810 |
| tanh | 0.753 | 0.805 | 0.783 | 0.799 | 0.849 |
| relu | 0.780 | 0.770 | 0.806 | 0.775 | 0.889 |
| gelu | 0.767 | 0.780 | 0.773 | 0.795 | 0.829 |
| sigmoid | 0.754 | 0.767 | 0.800 | 0.799 | 0.860 |

Table 6: **Full ablation studies for activation functions.** The values in the table represent the normalized validation loss for *Task*. For each setting, the experiments are repeated for multiple times. It can be observed that the model using $sin$ for both hidden and output layer achieves the best results.

