# OpenReview forum: "A Differentiable Physical Simulation Framework for Soft Robots on Multiple-Task Learning"
_ICLR.cc/2024/Conference — Submitted to ICLR 2024_

### Official Review · Reviewer_h9Cp · 2023-10-27

**Soundness:** 3 good
**Presentation:** 3 good
**Contribution:** 3 good
**Rating:** 6
**Confidence:** 3

**Summary:**

This paper presents a system to train controllers for soft robots in simulation. Key contributions are
(1) GPU-based, differentiable soft robot simulation
(2) design of a learning framework to use the simulation to train locomotion policies, including investigation of loss functions, activation functions for the neural network.

**Strengths:**

1. Design of a system for efficient learning soft robot locomotion skills.

2. Detailed analysis of different components of the system. Including gradient from the simulator, loss design and activation function.

**Weaknesses:**

Not much of technical contributions in terms of machine learning techniques. But hopefully the proposed system can be used to accelerate research in learning soft robot control research.

It is also not clear the importance of a GPU accelerated simulator, since the proposed system is not taking advantage of the potential of large number of parallel simulations.

**Questions:**

Question for the PPO set up. It only uses 8 processes and only collect small amount of data per iteration. Since the simulation is GPU accelerated, I wonder if it is possible scale these numbers up. As demonstrated in many prior work, scaling these numbers can make significant differences.

---

### Official Review · Reviewer_FNgh · 2023-10-30

**Soundness:** 1 poor
**Presentation:** 2 fair
**Contribution:** 2 fair
**Rating:** 3
**Confidence:** 4

**Summary:**

This paper presents multi-task learning for soft robot simulations using gradient-based control optimization, comparing it to reinforcement learning baselines. The differentiable simulation environment is implemented to support both mass-spring and material point method systems, and can be highly efficient through GPU parallelization.

**Strengths:**

An important challenge is being tackled for soft robotic simulation, where using these in practice is still far from feasible, and having a multi-task control framework would support the practical usability of soft robot simulations. The methodology in the paper is described clearly, and the supporting video helps explain the task descriptions in more detail, and is very much appreciated.

**Weaknesses:**

1. The simulation environment is reported to be interactive, what sort of framerates are considered as interactive, and how does this change for the problems with different complexity? Mentioning runtimes of the simulation would be beneficial.
2. It is indeed very impressive how such a simple MLP can learn various tasks for the presented robots, however, the question would be how this scales to more complex problems with complex robots. In the related works it would help to include how multi-task reinforcement learning has worked on the problem.
3. It is unclear what the novelty of this paper is, on the one hand, the GPU-accelerated simulation is presented as a contribution. In this case, a more detailed explanation of what is novel should be clear, which currently mainly is described by how previous work has been incorporated. For example, the friction model was explained on a high level with empirical data for results, yet more technical details to the implementation of friction and contact is missing, and more importantly, how it is different from previous work such as ChainQueen. A more thorough comparison with other simulators would then also be required if the simulation is part of the novelty, in both runtime and accuracy, and if applicable to real-world robots, some experimental results to validate sim-to-real should be added. On the other hand, if the simulation is a tool that was developed for the robot skill learning goal, a different variety of soft robots and tasks could have been tried, and more focus could have been put on this controlling part of the paper. The training of these networks, specifically for single-tasks, however, is similar to previous work such as ChainQueen or DiffPD. So a clearer definition in how these frameworks differ, would be much appreciated.
4. The validation set of tasks is only ever interpolating between task targets seen during training, does it make sense to also verify generalizability by extrapolating?
5. Proving that the gradients are not exploding or vanishing should be done at different timesteps, so it's clear that at earlier timesteps the gradient are similar to those at later timesteps. Right now saying what the range of the gradient norm is over all timesteps only shows there is no explosions/vanishing yet, although the range is -10 to 10 in log scale, which are quite smaller/big gradients already. But then again it's the sum of gradient norms, not the mean.
6. In Figure 8 caption, the robot design is not shown in the figure, and no actuator springs are marked.
7. A lot of English spelling mistakes that are hard to ignore should be fixed in the final version. Examples are:
 - Page 2: "... which enables a single neural network can be trained via..." grammatically 'can' should not be there.
 - Page 6: "... while the it usually converges"
 - Figure 5: "Iteartions"
 - Page 7: "... learning is the that the gradient"

**Questions:**

1. Physical parameter tuning was mentioned as future work, but since actuator placement has already been done with previous MPM papers, would this not be an easy feasible extension for the current paper? It would help solve the issue with training performance being degraded with improper robot designs.

---

### Official Review · Reviewer_ReY1 · 2023-10-31

**Soundness:** 3 good
**Presentation:** 3 good
**Contribution:** 3 good
**Rating:** 5
**Confidence:** 4

**Summary:**

The authors are presenting a framework for differentiable simulation for soft robots, which they use to teach a simulated soft robot a variety of tasks, such as moving, turning and jumping with a single neural network controller. The controller is learned by calculating the loss after several hundred steps of alternation of controller action followed by differentiable simulation and feedback. The task is specified to the robot controller in form of losses that describe the progress of the robot in a specific direction, or the maximum height obtained by the robot.

**Strengths:**

* The overall framework shows that the authors have a good understanding of the detailed technical issues involved in the physical simulation of soft robots and the challenges of learning the movement model.
* Very clear description of the various decisions made in the implementation.
* Extensive experimental studies, including a detailed ablation study.

**Weaknesses:**

* The proposed architecture is very specific to the type of the robots considered.  It is unclear what would be the pathway from this architecture to a physical robot.
* Many parts of the proposed architecture had been extensively engineered to achieve the objectives. To some degree the system appears to be halfway engineered, and only maybe halfway learned.
* The writeup for the contributions of the paper does not clearly outline what part of the architecture is new.

**Questions:**

* Please describe more clearly what new contributions are claimed, as the summary at the end of intro is not clear in this respect.
* It appears from the ablation study that some of the components of the work have a critical importance, although they are not highlighted by the paper - for instance, without the periodic signal the system is essentially not learning. It would be useful if we would learn more about why some of these components are so important.
* It is very unusual that in a neural network a sine activation function to lead to anything useful. Are the neurons in the architecture function in regimes that the sine activation function is activated several humps down???

---

### Official Review · Reviewer_5sx5 · 2023-11-01

**Soundness:** 3 good
**Presentation:** 3 good
**Contribution:** 3 good
**Rating:** 6
**Confidence:** 4

**Summary:**

* The GPU-accelerated high-performance differentiable physical simulation environment
    * Leverage DiffTaichi
    * Mass-spring systems
        * Change the rest-length of springs to generate forces
    * Material point methods
        * Use dynamic grid
        * Modify the Cauchy stress in the vertical direction of material space to apply this force
* The NN controller for soft robots on multiple locomotion tasks learning.
    * NN architecture is based on SIREN architecture, two linear layers with sin activation.
    * The gradients for training the network comes from the differentiable simulator.
* Multiple scenarios are tested, including running, jumping and crawling, rotating, juggle, dribble and shot. And several experiments are done to analyze the factors contributing to the differentiable physics based learning framework.
    * Ablation study on multiple components
    * Analysis of friction coefficient
    * Analysis of the gradients
    * Experiments of learning techniques
    * Compare the paper’s method with diffTaichi and PPO, where the paper shows better performance.

**Strengths:**

- A fair amount of experiments are done.
    - There are multiple ablation studies and each experiments are performed at least 3 times.
- Relatively complex cases (i.e., soft robots with complex shapes on multiple tasks) are shown in this paper.

**Weaknesses:**

- The paper uses multiple existing components, for example, diffTaichi as the differentiable simulator, and SIREN as the network architecture. I have some minor concerns regarding the novelty of this paper.
- Besides, I have some additional questions related to the experiments listed in the “Questions” section.

**Questions:**

- Questions for the ablation study:
    - For the setting of “remove optimizer”, does it mean the network weights were not updated? Then, why is the loss still changed, is it because different training data is used in different iteration?
    - For the batch size ablation, when different batch sizes are used, are the total amount of training samples different or same?
        - From the paper, “The network is trained with a batch size of 32. We train our models on NVidia RTX2060 GPU, the maximum training iteration is set to 10000, while the it usually converges in less than 5000 iterations within half an hour”
        - Does this mean larger batch size corresponds to more training samples (num_samples = batch_size * iterations?)?
    - The model trained with “full” still has a relatively large loss value - The loss only drops from 1 to 0.75. Is this mainly because of the regularization term?
- What’s the difference between [Difftaichi + your task loss] and your method (which also utilizes diffTaichi as the differentiable simulator)? Is the different in the SIREN network?
- In the discussion of friction model, the model performs best under “slip-0.4”. But for the other experiments (for example, the ablation study), seems like the setting of “sticky” is used. Why did you choose this environment for conducting all the other experiments?
- Different environment should have different friction coefficient, besides friction coefficient = 0.4, can the model also perform well on other coefficients?
- In gradient analysis, the range of 10^-10 to 10^10 seems very large already. Could you elaborate more on this?
- Proximal Policy Optimization (PPO) is published in 2017, are there any more recent methods that can be compared with this paper?
- For the “Periodic Signal” part in the input, is it a scalar?
- In Figure 12, I am a bit surprised that the network can still learn relatively well without the state vector input. “Full” converges to ~0.75, while “Full-SV” converges to ~0.78. Could you elaborate a bit more on why the network still works without knowing the system’s current states?
- The NN used in this paper only contains two linear layers, which is a relatively simple network within the field of machine learning. There are alternative network architectures, such as graph neural networks, that can more leverage the inherent structures of objects. Have you also considered the potential benefits of exploring other neural network architectures, although it may not be directly related to the novelty of this paper of using a differentiable simulator for training multi-task controller networks?

---

### Meta-Review · Area_Chair_gQ1w · 2023-12-10

**Metareview:**

This paper presents a framework for learning controllers for soft robots in simulation using differentiable physics and gradient-based optimization. The paper claims two main contributions: (1) a GPU-accelerated, high-performance, differentiable simulator for soft robots based on mass-spring systems and material point methods, and (2) a practical training module for soft robots to learn multiple locomotion tasks with a single neural network controller.


Strengths:

- The paper provides a comprehensive ablation study that validates the design choices made in the proposed method.


Weaknesses:

- The proposed architecture is very specific to the type of robots considered. It is unclear how it would scale or generalize to more complex robots or physical scenarios.

- Many parts of the proposed architecture are extensively engineered, and the system appears to be halfway engineered and halfway learned.

- Technical contributions in terms of machine learning techniques are not clear.

With many unresolved issues, the paper cannot be accepted in this form.

**Justification For Why Not Higher Score:**

Without resolving the major issues listed above, the paper cannot be accepted. Unfortunately, the authors did not provide rebuttals.

**Justification For Why Not Lower Score:**

N/A

---

### Decision · Program_Chairs · 2024-01-16

Reject